# Anti-Tumor Potential of *Gymnema sylvestre* Saponin Rich Fraction on In Vitro Breast Cancer Cell Lines and In Vivo Tumor-Bearing Mouse Models

**DOI:** 10.3390/antiox12010134

**Published:** 2023-01-05

**Authors:** Abhinav Raj Ghosh, Abdulrhman Alsayari, Alaa Hamed Habib, Shadma Wahab, Abhishek P. R. Nadig, Misbahuddin M. Rafeeq, Najat Binothman, Majidah Aljadani, Ibtesam S. Al-Dhuayan, Nouf K. Alaqeel, Mohammad Khalid, Kamsagara Linganna Krishna

**Affiliations:** 1Department of Pharmacology, JSS College of Pharmacy, JSS Academy of Higher Education and Research, Mysuru 570015, India; 2Department of Pharmacognosy, College of Pharmacy, King Khalid University, Abha 61421, Saudi Arabia; 3Complementary and Alternative Medicine Unit, King Khalid University, Abha 61421, Saudi Arabia; 4Department of Physiology, Faculty of Medicine, King Abdulaziz University, Jeddah 21589, Saudi Arabia; 5Department of Pharmacology, Faculty of Medicine, Rabigh, King Abdulaziz University, Jeddah 21589, Saudi Arabia; 6Department of Chemistry, College of Sciences and Arts, King Abdulaziz University, Rabigh 25732, Saudi Arabia; 7Department of Biology, College of Science, Imam Abdulrahman bin Faisal University, Dammam 31441, Saudi Arabia; 8Department of Pharmacognosy, College of Pharmacy, Prince Sattam Bin Abdilaziz University, Al-Kharj 11942, Saudi Arabia

**Keywords:** Gymnema sylvestre, anti-oxidant, cytotoxicity, Elrich ascites carcinoma, Dalton’s lymphoma ascites

## Abstract

*Gymnema sylvestre* (GS) is a perennial woody vine native to tropical Asia, China, the Arabian Peninsula, Africa and Australia. GS has been used as a medicinal plant with potential anti-microbial, anti-inflammatory and anti-oxidant properties. This study was conceptualized to evaluate the cytotoxicity potential of *Gymnema sylvestre* saponin rich fraction (GSSRF) on breast cancer cell lines (MCF-7 and MDA-MB-468) by SRB assay. The anti-tumor activity of GSSRF was assessed in tumor-bearing Elrich ascites carcinoma (EAC) and Dalton’s lymphoma ascites (DLA) mouse models. The anti-oxidant potential of GSSRF was assessed by DPPH radical scavenging assay. The acute toxicity of GSSRF was carried out according to OECD guideline 425. The yield of GSSRF was around 1.4% and the presence of saponin content in GSSRF was confirmed by qualitative and Fourier transform infrared spectroscopic (FTIR) analysis. The in vitro cytotoxic effects of GSSRF on breast cancer cell lines were promising and found to be dose-dependent. An acute toxicity study of GSSRF was found to be safe at 2000 mg/kg body weight. GSSRF treatment has shown a significant increase in the body weight and the life span of EAC-bearing mice in a dose-dependent manner when compared with the control group. In the solid tumor model, the doses of 100 and 200 mg/kg body weight per day have shown about 46.70% and 60.80% reduction in tumor weight and controlled the tumor weight until the 30th day when compared with the control group. The activity of GSSRF in both models was similar to the cisplatin, a standard anticancer agent used in the study. Together, these results open the door for detailed investigations of anti-tumor potentials of GSSRF in specific tumor models, mechanistic studies and clinical trials leading to promising novel therapeutics for cancer therapy.

## 1. Introduction

Cancer is a genetic disorder that results from any genetic or epigenetic alterations in the somatic cells [1]. Tumorigenesis is a multi-step process that transforms normal cells into malignant ones [2]. All around the world, the total number of estimated new cancer cases and deaths in 2022 was about 1.9 million, of which 609,360 were cancer deaths in the United States alone. Some plants, vegetables, herbs and spices used in folk and traditional medicines have become subjects of great interest for investigation to see whether they really possess anticancer properties that can be scientifically validated and recommended for possible therapeutic use against cancer [3,4].

*Gymnema sylvestre* (GS) is a perennial woody vine native to tropical Asia, China, the Arabian Peninsula, Africa and Australia. It has been used in Ayurvedic medicine, and the triterpene saponins (gymnemic acids), gymnema saponins and gurmarin have accounted for its anti-diabetic properties [5]. The leaves have also shown potential anti-arthritic activity due to triterpenoids, steroids and saponin glycosides. Other activities reported on the leaf extract were anti-microbial, anti-inflammatory, anti-hyperlipidemic, hepatoprotective and immunostimulatory [6]. Several triterpenoid saponins have been associated with a number of biological properties and cytotoxic potential, as well as apoptotic potential. Heredin, astragaloside and cucurbitacins are some of the notable examples of promising anti-cancer components [7]. Research on cytotoxic activity was carried out using this herb on HeLa cervical cancer cell lines [8], HT29 human colon adenocarcinoma cells [9], as well as the human osteosarcoma cell line MG63 [10]. These studies suggest the possible scope of this herb as an anti-cancer agent. Based on the available literature, the present study was conceived to evaluate the cytotoxic potential of GS saponin rich fraction (GSSRF) in in vitro and in vivo models. 

## 2. Materials and Methods

### 2.1. Preparation of GSSRF

The powdered leaves of this plant were obtained from Udhagamandalam, Tamil Nadu and were defatted by petroleum ether for 3 h at 40 °C. After filtering the petroleum ether, the sample was extracted with methanol for 3 h with mild heating. The methanol extract was concentrated and re-extracted with methanol and acetone (1:5 *v*/*v*). The precipitate obtained was dried under vacuum; it turned to a whitish amorphous powder after complete drying [8].

### 2.2. Animals

The study was performed on Swiss albino mice (SAM) of either gender weighing 25 to 35 g. They were supplied by Biogen Laboratory Animal Facility, Bangalore. All protocols related to animal handing were carried as per CPCSEA guidelines issued by the Institutional Animal Ethics Committee (IAEC), JSS College of Pharmacy, Mysore, Karnataka. The studies conducted were approved by the Institutional Animal Ethics Committee, JSS College of Pharmacy, Mysore, Karnataka (Approval no: IAEC/JSSCPM/331/2019).

### 2.3. Cell Lines

The Elrich ascites carcinoma (EAC) and Dalton’s lymphoma ascites (DLA) cells were originally obtained from the Centre of Excellence in Molecular Biology and Regenerative Medicine (CEMR), Department of Biochemistry, JSS Medical College, JSS Academy of Higher Education and Research, Mysuru and were maintained in our animal house by serial intraperitoneal (i.p.) transplantation in SAM. The EAC cells used for the study were collected from the i.p. space of mice after 12–14 days of cell inoculation. Human breast adenocarcinoma—double-positive (MCF-7) and triple-negative cell lines (MDA-MB-468)—were also procured from the CEMR facility, Mysuru. Cells were routinely grown in culture flasks (Tarsons Products Pvt. Ltd., West Bengal, India) containing Dulbecco’s modified eagle medium (DMEM, from Gibco, WA, USA) supplemented with 10% fetal calf serum (FCS, from Sigma-Aldrich, St. Louis, MO, USA) and 0.5 mg/mL penicillin-streptomycin (Pen-Strep, from Gibco, Waltham, MA, USA) at 37 °C in a CO_2_ incubator (5% CO_2_) [11].

### 2.4. Test Drug Preparation

For in vitro studies, a stock solution was prepared by solubilizing the extract in 0.2% *v*/*v* dimethyl sulfoxide and the sample was further diluted to 
prepare concentrations of 1.56, 3.12, 6.25, 12.50, 25.00, 50.00 and 100.00 µg/mL. Diallyl disulfide was used as the standard against breast cancer cell 
lines. For in vivo studies, extracts were solubilized in distilled water and administered orally with a volume based on the animal weight.

### 2.5. Standardization of GSSR

The purified saponins were subjected to structural elucidation by Fourier transform infrared spectroscopic analysis (FTIR) [12] and chemical tests (foam test) for determination of saponin constituent [13].

### 2.6. DPPH Radical Scavenging Assay

The anti-oxidant activity of GSSRF was determined using DPPH assay at five different concentrations 20, 40, 60, 80 and 100 µg/mL [14]. 

### 2.7. In Vitro Cytotoxic Activity on MCF-7 and MDA-MB-468 Cell Lines

In vitro cytotoxicity of GSSRF was performed on MCF-7 and MDA-MB-468 cells using sulforhodamine B (SRB) assay in 96-well plates. The cells were treated with increasing concentration ranging from 1.56, 3.12, 6.25, 12.50, 25.00, 50.00 and 100.00 µg/mL of GSSRF along with a positive control (diallyl disulfide-1 mM) for 24 and 48 h and cell viability determined using SRB assay [15,16].

### 2.8. Acute Toxicity Studies

The acute toxicity study of GSSRF was conducted as per OECD 425 guidelines at a dose level of 2000 mg/kg body weight in female SAM. Dosing was carried out as per OECD 423 guidelines [17,18]. 

### 2.9. EAC-Induced Liquid Tumor Model

The induction and propagation of tumors were performed according to the procedures of Jagetia and Rao [19,20]. A known number of viable EAC cells (2.5 million cells/mice) were injected intraperitoneally in aseptic conditions. Tumors ere inoculated on day zero. Twenty-four hours after EAC cell inoculation (day one), the tumor-bearing animals were randomly divided into different groups, as shown in Table 1. Six animals were included in the normal group, whereas all other groups have twelve animals each; cisplatin was used as the reference standard (Table 1). The mean survival time (MST) was noted from the day of the EAC cell line inoculation for the control and treatment groups and % increase in life span (%ILS) was calculated by using the MST of the control and treatment groups.

### 2.10. DLA-Induced Solid Tumor Model

DLA cells were obtained from Swiss albino mice after 15 days of inoculation. The fluid was aspirated and tested for microbial contamination [21]. The viability of the cells and the cell number were assessed by the trypan blue exclusion test. The cell number was adjusted with saline to obtain a cell suspension of one million cells per ml. A total of 0.1 mL of the cell suspension was injected intramuscularly into the right hind limb of male SAM to obtain a solid tumor. Treatments were started 24 h after tumor inoculation. The tumor-bearing animals were randomly divided into different groups, as shown in Table 2. Six animals were included in all groups and cisplatin was used as the reference standard (Table 2).

### 2.11. Histopathology 

In many solid tumors, lymph nodes are the first sites of cancer spread [22]. According to emerging research, the lymph node microenvironment provides hospitable soil for cancer seeding and growth. Furthermore, the presence of tumor cells in regional or sentinel lymph nodes (SLN) has significant clinical importance since it is linked with disease progression and poor prognosis and frequently influences therapy selection [23]. The lymph node of DLA-induced solid tumor was extirpated and maintained in 10% formalin solution and was sent to Genespy research services, Mysuru for hematoxylin and eosin staining.

### 2.12. Statistical Analysis

Data were analyzed by one-way ANOVA using GraphPad Prism version 8.0.2 for Windows, GraphPad Software, San Diego, CA, USA, followed by Tukey’s post hoc test. A value of *p* < 0.05 was considered statically significant.

## 3. Results

### 3.1. Percentage Yield of GSSRF

The percentage yield of GSSRF was based on the weight crude methanol extract. The yield of GSSRF was found to be 1.4% as sown in Table 3.

### 3.2. Chemical Test and Spectroscopic Data

The presence of saponins was confirmed by performing the foam test, which was positive. GSSRF showed IR spectra at 3529.86, 2834.29, 1643.41, 1460.16 and 1105.25 per cm (Figure 1).

### 3.3. In Vitro Anti-Oxidant Study by DPPH Scavenging Assay

DPPH assay was performed at 20, 40, 60, 80 and 100 μg/mL concentrations of GSSRF. The percentage DPPH scavenging by GSSRF was found to be 9.05, 24.99, 34.96, 40.21 and 73.73, respectively. The percentage scavenging increased in a dose-dependent manner and the maximum scavenging was observed at the 100 μg/mL dose. The IC_50_ value of GSSRF was found to be 85.02 ± 0.52 µg/mL (Figure 2).

### 3.4. In Vitro Cytotoxic Effects on MCF-7 and MDA-MB-468 Cell Lines

The saponin rich fraction was found to be relatively safe on Vero cell lines based on the studies performed by Khanna et al. [7]. Percentage cytotoxicity on cancer cell lines (MCF-7 and MDA-MB-468) was calculated at 24 and 48 h of incubation with GSSRF. The concentrations of extracts 1.56, 3.12, 6.25, 12.50, 25.00, 50.00 and 100.00 µg/mL showed cytotoxicity on both cell lines, with the highest cytotoxicity observed at 100 µg/mL. At 24 h, the IC_50_ value of GSSRF on MCF-7 and MDA-MB-468 cells was found to be 63.77 ± 0.23 μg/mL and 103.14 ± 1.05 μg/mL, respectively. When the incubation was extended for 48 h, the MCF-7 and MDA-MB-468 cells exhibited the IC_50_ value of 114.01 ± 0.13 μg/mL and 135.33 ± 2.40, respectively. In contrast, GSSRF was recorded with higher cytotoxicity against breast cancer cell line MCF-7 at 24 and 48 h (Table 4).

### 3.5. Acute Toxicity Study 

No mortality was observed at 2000 mg/kg body weight and was found to be safe. No signs of neurological and behavioral toxicity were observed at this dose level using Irwin’s observation test. Dosages of 1/80th, 1/40th and 1/20th of the safe dose (2000 mg/kg), i.e., 25, 50 and 100 mg/kg, were selected for the in vivo anti-tumor study.

### 3.6. EAC-Induced Liquid Tumor Model 

#### 3.6.1. Body Weight Changes

A significant body weight gain of EAC-inoculated control mice was observed at day 15 compared with day 0, with a maximum gain (41.81 ± 0.44). Compared with the control group, the body weight (32.55 ± 0.65, 30.69 ± 0.95 and 29.79 ± 1.07) had significantly (*p* < 0.05) reduced with GSSRF treatment of 25, 50 and 100 mg/kg, respectively. The cisplatin at 3.5 mg/kg reduced body weight significantly (*p* < 0.05) when compared with the control (24.43 ± 0.51) (Table 5). The percentage increase in body weight was also calculated and there was a significant increase in the control group, which was around 98.81% on the 15th day of tumor inoculation. GSSRF administration did not reduce the body weight until the 12th day and the increase in body weights were 53.61%, 48.24% and 47.97% for 25, 50 and 100 mg/kg doses, respectively. The body weight decreased on the 15th day with the percentage of body weight being 52.67%, 41.64% and 40.39% for each dose, respectively (Table 5). The standard drug cisplatin (3.5 mg/kg) showed a slight increase of 25.54% until the 9th day, followed by a decrease of 13.68% until the 15th day.

#### 3.6.2. Survival Study

The control animals showed a significant (*p* < 0.05) decrease in mean survival time (MST) when compared with the standard group, whereas oral administration of GSSRF exhibited a dose-dependent significant (*p* < 0.05) increase in MST when compared with the control group (Table 6). The percentage increase in life span (% ILS) was also calculated. Compared with the control group, the % ILS increased (28.37%, 41.98% and 70.37%) with GSSRF treatment of 25, 50 and 100 mg/kg, respectively (Table 6) (Figure 3).

#### 3.6.3. Hematological profile

In the EAC-induced control mice (4.12 ± 0.37), a substantial decline was found in overall RBC counts compared with the normal mice (6.33 ± 0.15). The GSSRF at 25, 50 and 100 mg/kg showed a significant (*p* < 0.05) increase in RBC counts (4.39 ± 0.34), (5.26 ± 0.26) and (5.71 ± 0.46) when compared with the control group (Table 7). Standard cisplatin at 3.5 mg/kg significantly (*p* < 0.05) reversed the decrease in the total RBC count (6.66 ± 0.15) compared with the control group. 

In the EAC control mice (52.51 ± 1.02), a significant (*p* < 0.05) increase in WBC counts was observed compared with the normal (10.65 ± 0.34) mice. The GSSRF reported a significant (*p* < 0.05) decrease (46.72 ± 17.03), (25.55 ± 4.84) and (20.71 ± 3.28) at 25, 50 and 100 mg/kg. The standard drug cisplatin reduced the increased WBC count significantly (11.97 ± 3.32) when compared with the control (Table 7). 

In the EAC control mice (5.06 ± 0.09), a significant (*p* < 0.05) decrease in the hemoglobin level was observed compared with the normal mice (11.30 ± 0.26). The GSSRF at 25, 50 and 100 mg/kg showed a significant (*p* < 0.05) increase in the hemoglobin level (7.07 ± 0.58), (9.20 ± 0.33) and (9.88 ±0.91) when compared with the control. Standard cisplatin at 3.5 mg/kg significantly (*p* < 0.05) reversed the decrease in the total hemoglobin level (10.47 ± 0.24) (Table 7).

### 3.7. DLA-Induced Solid Tumor Model

#### 3.7.1. Tumor Volume

The tumor volume in the control animals was significantly (*p* < 0.05) increased, with a maximum (0.85 ± 0.01), on day 30 when compared with day 0. The treatment with GSSRF at 100 and 200 mg/kg significantly (*p* < 0.05) reduced the tumor volume (0.51 ± 0.01) and (0.48 ± 0.01) when compared with the control; a more significant reduction was seen at the dose of 200 mg/kg. The standard cisplatin reduced the tumor volume (0.42 ± 0.01) in comparison with the control (Figure 4 and Figure 5).

#### 3.7.2. Tumor Weight

The tumor weight was significantly (*p* < 0.05) increased in the DLA-inoculated control mice, with a maximum gain of (7.86 ± 0.25) on day 30. The treatment with GSSRF at 100 and 200 mg/kg significantly (*p* < 0.05) reduced the tumor weight (4.19 ± 0.34) and (3.08 ± 0.04), with the percentage reduction in tumor growth at 46.70% and 60.80% when compared with the control group. The maximum tumor weight reduction was observed at 200 mg/kg. Standard cisplatin also significantly (*p* < 0.05) reduced the tumor weight (0.92 ± 0.17) when compared with the control group (88.32%) (Figure 6).

### 3.8. Histopathology Studies

In the control group, necrotic changes in lymph nodes and loss of normal architecture were observed and eosinophilic areas of necrosis with inflammatory cells were observed. Diffuse infiltrate of lymphoid cells was also visible (Figure 7A). The standard group exhibited a normal histology of lymph nodes. Normal lymphocytes with follicles in the paracortical region seen beneath the capsule and interfollicular region rich in lymphocytes were observed (Figure 7B), whereas the group treated with GSSRF at 100 mg/kg exhibited a normal nodal architecture and a pale germinal center of the follicle remained to be normal (Figure 7C). Similarly, the group treated with GSSRF at 200mg/kg revealed a lymph node, showing the effacement of the lymph node architecture. The architecture was effaced by the diffuse infiltrate of lymphoid cells. In some places, interfollicular or sinusoidal was observed and necrosis seen with eosinophilic cellular debris and the presence of erythrophagocytosis was also observed (Figure 7D).

## 4. Discussion 

The present study evaluated the anti-tumor efficacy of the leaves of *Gymnema saponins* (GS) in different cancer cell lines and transplantable tumor-bearing mice. This study is based on the previous anti-cancer studies performed on the leaves [24]. The presence of oleanane (gymnemic acids and gymnema saponins) and dammarane classes (gymnemasides) of triterpenes were previously identified to be the major saponin components of this plant [11].

### 4.1. In Vitro Studies of GS

GSSRF has shown significant cytotoxic potential SRB assays on breast cancer cell lines against proliferation and growth of the cells. The possible mechanisms for the potential cytotoxic activity may involve the apoptotic induction in ER-negative breast cancer cells through the mitochondrial pathways of mitochondrial transmembrane potential dissipation, the activation of caspase-3 and caspase-9, the cytosolic release of cytochrome c, the reduction in reactive oxygen species (ROS), the arrest of the cell cycle, the protective mechanism against oxidative damage and the up-regulation of p53 (Figure 8) [25,26]. MCF-7 and MDA-MB-468 cell lines were used to assess the long-term cytotoxic potential (24 and 48 h) using SRB assay and, in both cell lines, the dose of 100 µg/mL was found to be most toxic after 24 h and 48 h exposure. Previous studies carried out using this fraction found that this fraction was not toxic to normal cells [8]. 

### 4.2. In Vivo Studies of GSSRF

Acute toxicity studies were performed according to OECD guidelines 425 and the dose of 2000 mg/kg was found to be safe based on Irwin’s observation test for behavioral and physiological functions. The GSSRF of 25, 50 and 100 mg/kg were tested on an EAC-induced liquid tumor model in mice. The maximum increase in body weight was observed in the control and the maximum growth reduction was seen in the cisplatin-treated group. The three doses of GSSRF reduced the body weight in a dose-dependent manner. The maximum body weight reduction was observed at the dose of 100 mg/kg. Due to its anti-oxidant and cytotoxic property, we believe it reduced both the inflammation in the peritoneal cavity as well as the infiltration of proliferative cells, in turn reducing the body weight [27]. In hematological parameters, the EAC-induced mice showed an increased WBC count and decreased RBC count and Hb level. Due to the presence of inflammatory mediators, there is a tendency of an increased proliferation of WBC cells in the body of cancer animals. Due to the myelosuppression or hemolysis, RBC synthesis in the body is decreased with the presence of immature RBC cells in the blood. This in turn leads to low Hb content. The GSSRF doses of 25, 50 and 100 mg/kg showed a gradual improvement in hematological parameters. The cisplatin-treated group showed a decrease in the number of WBC cells and an improvement in RBC count and Hb levels. Generally, EAC induces an increase in neutrophils due to the acute inflammatory response. The improvement in hematological parameters signifies the immunostimulant potential and the reversal of hematological parameters revealed the protective action on the hematopoietic system without the induction of myelotoxicity [28]. After 15 days of treatment, the life span assessment was carried out and GSSRF 100 mg/kg increased the % ILS by 70.37% in comparison with 25 mg/kg (28.37%) and 50 mg/kg (41.997%). In comparison, the cisplatin-treated group showed 96.29% ILS when compared with the control group. This may be due to the anti-tumor property of GSSRF in prolonging the life span of animals in comparison with standard cisplatin [29]. The GSSRF was also tested against the DLA-induced solid tumor model in mice. The anti-tumor activity was tested at two doses of 100 and 200 mg/kg. In the control animals, more tumor volume and an increased tumor weight were observed. GSSRF at 100 and 200 mg/kg doses controlled the tumor volume, and tumor weight reduction was observed when compared with the control group. In the cisplatin-treated group, there was more reduction in tumor volume and tumor weight. These results demonstrate that, this plant may be a possible source of natural anti-oxidants and may be of great significance in the prevention or slowing of degenerative diseases such as cancer linked with oxidative stresses [30].

## 5. Conclusions

From the results obtained, it may be concluded that the selected potent fraction GSSRF showed significant cytotoxic activity, which was evident from the in vitro cytotoxicity assays on double-positive (MCF-7) and triple-negative (MDA-MB-468) cell lines. The in vitro result was well supported by the in vivo activity against the liquid tumor and the solid tumor mouse models. Thus, this study is an initial step in the identification of a novel and selective herbal anti-tumor agent, which may be devoid of many of the adverse effects of anti-cancer chemotherapy. However, at this point of study, the exact mechanism cannot be determined and further studies are required to assess the anti-tumor mechanism.

## Figures and Tables

**Figure 1 antioxidants-12-00134-f001:**
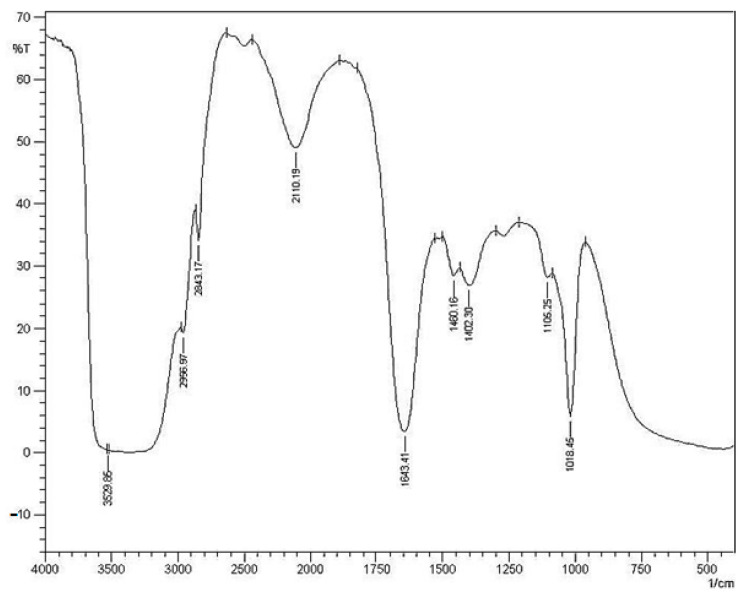
IR Spectroscopy of methanol soluble fraction of GSSRF.

**Figure 2 antioxidants-12-00134-f002:**
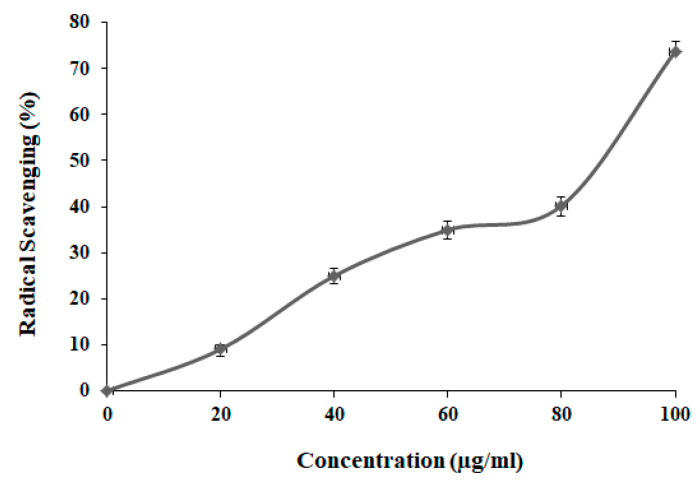
DPPH scavenging assay of GSSRF.

**Figure 3 antioxidants-12-00134-f003:**
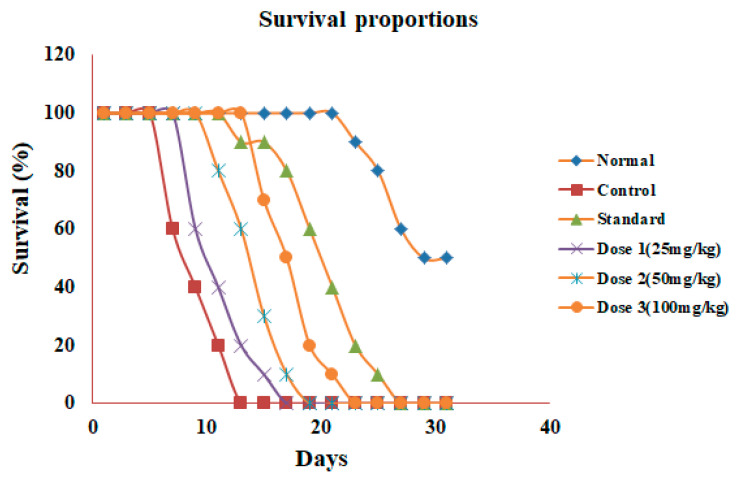
Effect of GSSRF on EAC-induced liquid tumor model (mean survival time).

**Figure 4 antioxidants-12-00134-f004:**
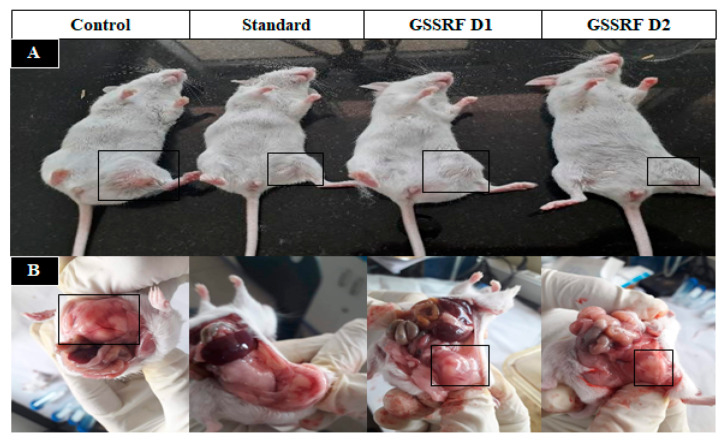
Effect of GSSRF on DLA-induced solid tumor model. (**A**) Physical appearance. (**B**) Tumor volume of control, standard, GSSRF D1- and GSSRFD2-treated tumor mice.

**Figure 5 antioxidants-12-00134-f005:**
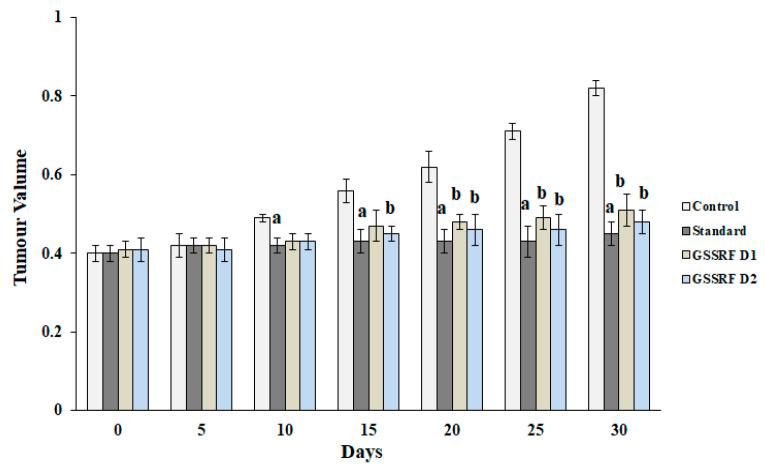
Effect of GSSRF on DLA-induced solid tumor model (tumor volume). Values are expressed as mean ± SEM, *n* = 6. *p* < 0.05 ^a^ significant when compared with control vs. standard group. *p* < 0.05 ^b^ significant when compared with control vs. treatment group.

**Figure 6 antioxidants-12-00134-f006:**
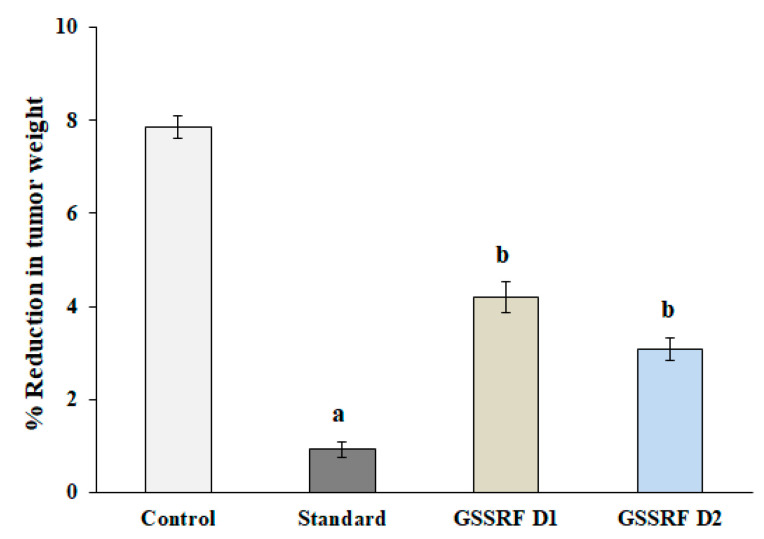
Effect of GSSRF on DLA-induced solid tumor model (tumor weight). Values are expressed as mean ± SEM, *n* = 6. *p* < 0.05 ^a^ significant when compared with control vs. standard group. *p* < 0.05 ^b^ significant when compared with control vs. treatment group.

**Figure 7 antioxidants-12-00134-f007:**
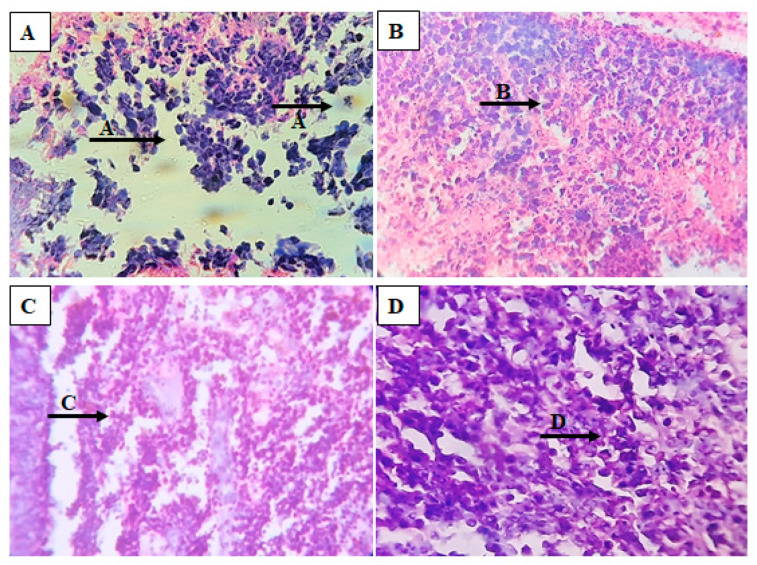
Hematoxylin and eosin staining of tumor sections of DLA-induced mice. (**A**): control, (**B**): standard, (**C**): GSSRF (100 mg/kg) and (**D**): GSSRF (200 mg/kg).

**Figure 8 antioxidants-12-00134-f008:**
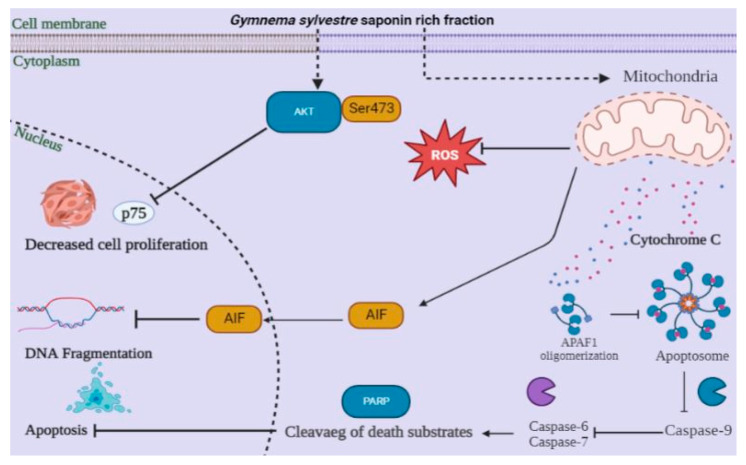
Anti-cancer activity and possible mechanism of action of *Gymnema sylvestre* saponin rich fraction.

**Table 1 antioxidants-12-00134-t001:** Grouping and treatment schedule of experimental animals in EAC model.

Group	Treatment	Evaluation
Normal	Distilled water (vehicle) *p.o.* 15 days.	Percentage increases in body weight on 3rd, 6th, 9th, 12th and 15th day. MST and % ILS. Hematological parameters (hemoglobin, RBC and WBC).
Control	Vehicle for *p.o.* 15 days + EAC cells.	
Standard	Cisplatin (3.5 mg/kg) *i.p*. twice a week + EAC cells.	
GSSRF D1	25 mg/kg *p.o*. for 15 days + EAC cells.	
GSSRF D2	50 mg/kg *p.o*. for 15 days + EAC cells.	
GSSRF D3	100 mg/kg *p.o*. for 15 days + EAC cells.	

*n* = 6 for normal group, *n* = 12 for treatment groups. MST—mean survival time; % ILS—increase in life span; EAC—Elrich ascites carcinoma; *i.p*.—intraperitoneal; *p.o*.—per oral.

**Table 2 antioxidants-12-00134-t002:** Grouping and treatment schedule of experimental animals in DLA model.

Group	Treatment	Evaluation
Normal	Distilled water (vehicle) *p.o*. for 30 days	Tumor volume on every 5th day. Tumor weight on 30th day. Histopathology
Control	Vehicle *p.o*. for 30 days + DLA cells	
Standard	Cisplatin (3.5 mg/kg) *i.p*. twice a week for 30 days + DLA cells.	
GSSRF D1	100 mg/kg *p.o*. for 30 days + DLA cells	
GSSRF D 2	200 mg/kg *p.o*. for 30 days + DLA cells	

*n* = 6. DLA—Dalton’s lymphoma ascites; *i.p.*—intraperitoneal; *p.o.*—per oral.

**Table 3 antioxidants-12-00134-t003:** The percentage yield of GSSRF.

Parts of Plant	% Yield *w*/*w*
Leaves of GS.	1.4

**Table 4 antioxidants-12-00134-t004:** Percentage cytotoxicity of GSSRF on MCF-7 and MDA-MB-468 cell lines.

Concentration (μg/mL)	% Cytotoxicity on MCF-7 Cell Line		% Cytotoxicity on MDA-MB-468 Cell Line	
	24 h	48 h	24 h	48 h
Vehicle control	18.40 ± 0.27	23.64 ± 0.08	18.50 ± 0.71	30.86 ± 1.17
DADS	67.50 ± 0.77	46.36 ± 1.22	67.77 ± 1.23	45.87 ± 1.79
1.56 µg/mL	08.35 ± 0.18	02.76 ± 0.16	06.25 ± 1.18	−4.97 ± 0.39
3.12 µg/mL	21.82 ± 0.13	11.15 ± 0.11	14.71 ± 1.09	12.05 ± 0.74
6.25 µg/mL	32.09 ± 0.29	18.07 ± 0.08	18.05 ± 0.43	15.21 ± 0.61
12.50 µg/mL	35.98 ± 0.41	20.43 ± 0.25	18.54 ± 1.07	16.73 ± 0.29
25.00 µg/mL	37.26 ± 0.46	25.36 ± 0.39	21.61 ± 0.62	18.89 ± 1.22
50.00 µg/mL	40.03 ± 0.09	27.81 ± 0.11	31.92 ± 0.78	21.51 ± 0.62
100.0 µg/mL	66.49 ± 0.11	43.82 ± 0.14	47.81 ± 0.59	38.52 ± 0.94
IC_50_ (µg/mL)	63.77 ± 0.23	114.01 ± 0.13	103.14 ± 1.05	135.33 ± 2.40

Values are expressed as Mean ± SEM, *n* = 3.

**Table 5 antioxidants-12-00134-t005:** Effect of GSSRF on EAC-induced liquid tumor model (body weight changes and percentage increase in body weight).

Groups	Body Weight Changes in gm (% Increase in Weight)					
	Day 0	Day 3	Day 6	Day 9	Day 12	Day 15
Normal	20.97 ± 0.81	21.68 ± 0.99 (3.38)	22.93 ± 1.22 (9.34)	23.86 ± 1.11 (13.78)	25.01 ± 1.06 (19.26)	25.68 ± 0.82 (22.46)
Control	21.03 ± 0.52	25.50 ± 0.43 ^a^ (21.25)	33.56 ± 0.39 ^a^ (59.58)	37.11 ± 0.43 ^a^ (76.46)	39.28 ± 0.29 ^a^ (86.78)	41.81 ± 0.44 ^a^ (98.81)
Standard	21.49 ± 0.55	23.70 ± 0.55 ^b^ (10.28)	24.95 ± 0.63 ^b^ (16.10)	26.98 ± 0.63 ^b^ (25.54)	26.00 ± 0.63 ^b^ (20.98)	24.43 ± 0.51 ^b^ (13.68)
GSSRF D1	21.32 ± 0.46	23.03 ± 0.45 ^c^ (8.02)	25.16 ± 0.42 ^c^ (18.01)	30.28 ± 0.43 ^c^ (42.02)	32.75 ± 0.66 ^c^ (53.61)	32.55 ± 0.65 ^c^ (52.67)
GSSRF D2	21.66 ± 0.58	23.62 ± 0.49 ^c^ (9.01)	25.75 ± 0.54 ^c^ (18.84)	29.90 ± 0.58 ^c^ (37.99)	32.12 ± 0.94 ^c^ (48.24)	30.69 ± 0.95 ^c^ (41.64)
GSSRF D3	21.22 ± 0.41	23.45 ± 046 ^c^ (10.50)	27.18 ± 0.50 ^c^ (28.08)	30.09 ± 0.61 ^c^ (41.80)	31.40 ± 1.09 ^c^ (47.97)	29.79 ± 1.08 ^c^ (40.39)

Values are expressed as mean ± SEM, *n* = 6. *p* < 0.05 ^a^ significant when compared with control vs. normal group. *p* < 0.05 ^b^ significant when compared with control vs. standard group. *p* < 0.05 ^c^ significant when compared with control vs. treatment group.

**Table 6 antioxidants-12-00134-t006:** Effect of GSSRF on EAC-induced liquid tumor model (mean survival time and percentage increase in life span).

Groups	Mean Survival Time (Days)	% Increase in Life Span
Control	13.50 ± 0.43	—
Standard (cisplatin 3.5 mg/kg)	26.50 ± 0.56 ^a^	96.29
GSSRF D1 (25 mg/kg)	17.33 ± 0.42 ^b^	28.37
GSSRF D2 (50 mg/kg)	19.17 ± 0.91 ^b^	41.98
GSSRF D3 (100 mg/kg)	23.00 ± 1.21 ^b^	70.37

Values are expressed as mean ± SEM, *n* = 6. *p* < 0.05 ^a^ significant when compared with control vs. normal group. *p* < 0.05 ^b^ significant when compared with control vs. standard group.

**Table 7 antioxidants-12-00134-t007:** Effect of GSSRF in EAC-inoculated mice (hematological parameters).

Groups	RBC Count (10^6^ Cells/mm)	WBC Count (1000 Cells/mm)	Hb Content (gm %)
Normal	6.33 ± 0.15	10.65 ± 0.34	11.30 ± 0.26
Control	3.03 ± 0.31 ^a^	52.51 ± 1.02 ^a^	05.06 ± 0.09 ^a^
Standard	6.66 ± 0.16 ^b^	11.97 ± 3.32 ^b^	10.47 ± 0.24 ^b^
GSSRF D1	4.39 ± 0.34	46.72 ± 1.03 ^c^	07.07 ± 0.58 ^c^
GSSRF D2	5.26 ± 0.26 ^c^	25.51 ± 4.84 ^c^	09.20 ± 0.33 ^c^
GSSRF D3	5.59 ± 0.19 ^c^	20.71 ± 3.28 ^c^	09.88 ± 0.91 ^c^

Values are expressed as mean ± SEM, *n* = 6. *p* < 0.05 ^a^ significant when compared with control vs. normal group. *p* < 0.05 ^b^ significant when compared with control vs. standard group. *p* < 0.05 ^c^ significant when compared with control vs. treatment group.

## Data Availability

All the data are available within the article.

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
