# Peer review of "Anti-Tumor Potential of Gymnema sylvestre Saponin Rich Fraction on In Vitro Breast Cancer Cell Lines and In Vivo Tumor-Bearing Mouse Models"

_antioxidants, 2023, doi:10.3390/antiox12010134_

Round 1

Reviewer 1 Report

The authors did a good job overall, investigating both the in vivo and in vitro properties of the Gymnema sylvestre extract. The project is interesting but corrections must be made on table 2 which is difficult to understand, especially for a better dissemination of the research. Probably due to a file formatting problem. However, the results are encouraging and I hope for more stringent studies on the properties of the leaf extract.

Reviewer 2 Report

This paper is potentially interesting, and may be clinically important for the future management of cancers. I have the following comments about the Ms.

<Title>

‘in-vitro breast cancer cell lines’ is okay, but the following one, ‘in-vivo tumor’ is not clear. My question is whether this ‘in-vivo tumor’ is also breast cancer or not. Or, something else? This point needs to be clarified within this title.

<Abstract>

Seeing the methods or results, it seems that this study compared the effect of cisplatin (as ‘Standard’) with that of GSSRF. This point is very important to evaluate the effect of GSSRF on cancers. But nowhere is found about this point in the abstract. This point should be described in the abstract. This is particularly important since this allows the readers to evaluate GSSRF more in detail in regard to cancers on their own.

<Materials and Methods>

#2.9 EAC-induced liquid tumor model

-line 5, ‘and treated treatment’: I could not understand this. This may be ‘and treatment’ as in 2.10.

-Table 1:

*In each of Normal to GSSRF D3, it is not clear how many mice were included. I believe it could not have been just one, though.

*In the 2nd row (Control), it says ‘++EAC cells’. Two +s mean something? After all, I could not understand what + means in this table. It means ‘intraperitoneal injection of EAC cells’?

*Control: It says here, according to the pdf I downloaded, ‘Cisplatin prepared in distilled water’. I could not understand this method. If cisplatin had been administered in some way or other, it could not have been controls. Or, the authors wanted to compare the effect of cisplatin with that of GSSRF??? Seeing Table 2, something of ‘cisplatin’ belongs to ‘Standard’. This is what the authors want to mean in Table 1 as well???

*duration of MST and hematological parameters were noted: ‘Noted’ what? What kind of hematological parameters were assessed?

#2.10 DLA-induced solid tumor model (Strictly speaking, ‘DLA-’)

*15 days old cell transplanted Swiss albino mouse: I could not understand this phrase. It is ‘15 day-old cell-transplanted Swiss albino mouse/mice’?

*Viability of cells and…was assessed >>> Viability of cells and…were assessed

*As in Table 1, it is not clear how many mice were allotted to each group.

*Seeing this Table 2, ‘Standard’, as well as that  in Table 1, may have been the group treated with cisplatin.

<Results>

#Many tables and figures are used to show the results of this study. To show them in a flawless way is definitely the responsibility of the authors.

#3.8 Histopathology studies

This result is completely impossible to understand. It says, throughout the Ms, ‘DLA-induced solid tumor model’, right? And actually, according to the materials and methods, DLA cells were injected intramuscularly to make a ‘solid’ tumor there. Then, why are the photos shown here from lymph nodes??? Do the authors want to show LN metastasis of the tumor??? Why don’t you show the solid tumor itself, which must have been located intramuscularly??? And one more: Figure 6 means nothing. There is no qualitative difference between A, B, C, and D. Something like necrosis could be visible, but that’s all. Why is it possible to say something with this kind of photos that are from deranged, artifactual, and crushed materials???

<Others>

Generally speaking, the Ms is written well with a good command of English, but there are many points that are grammatically/stylistically questionable, probably careless, or hard-to-understand expressions throughout the Ms. I guess this paper may be the one by English-speaking natives; but since an English paper must be read worldwide, I mean, by English-speaking as well as English-non-speaking readers, it needs to be as easily understandable and yet scientifically correct as possible. The Ms needs to be proofread again and again.

Ex (Only from the Abstract)

-‘by in-vitro breast cancer cell lines’ (title) vs ‘by in-vitro on breast cancer cell lines’ (abstract, line 4): These two are hard to understand. Are these right phrases? I guess ‘on in-vitro breast cancer cell lines’ is the right one for both???

-Abstract, line 3: ‘, anti-oxidant properties’ >>> ‘, and anti-oxidant properties’

-Abstract, line 5: ‘in-vivo efficacy in tumor bearing’ >>> ‘its in-vivo efficacy in tumors bearing’? And one more: What kind of tumors is indicated here? Breast cancer? Just carcinomas or lymphomas?

-Abstract, line 5, 6: …Carcinoma-induced? …Ascites-induced? (May be more easily understandable)

-Abstract, line 8, FTIR: What is FTIR? Or this happens to be FT-IR as in the Materials and Methods? Since an abstract should be self-contained, this abbreviation should be spelled out.

-Abstract, line 9: ‘The anti-oxidant and in-vitro toxicity on cancer cell lines effects of GSSRF were’ >>> ‘The anti-oxidant and in-vitro toxic effects of GSSRF on cancer cell lines were’???

-Abstract, line 9: dose-dependent?

-Abstract, line 10, ‘2000mg/kg’: Here, as well as somewhere else in the abstract, the unit ‘mg/kg’ is used. Is this okay??? If it represents the dose of medicine taken in in-vivo studies, it should be like ‘mg/kg/day’???

-Abstract, line 10-11: This sentence is a little difficult to understand (at least for me, though). ‘the dose of…showed maximum decline in the tumor-induced increase in body weight…in comparison with controls’???

-Abstract, line 14: ‘with the control’ or ‘with controls’ or ‘with the controls’???

-Abstract, line 15: ‘these results opens’ >>> ‘these results open’

-Abstract, keywords or text: ‘anti-oxidant’ or ‘anti-oxidant’?

-etc, etc, etc throughout the Ms

Reviewer 3 Report

To authors:

The study by Ghosh et al about the medicinal plant Gymnema sylvestre (GS) and it´s potential anti-microbial, anti-inflammatory, anti-oxidant properties. Here, the authors evaluated the antitumor poten- tials of Gymnema sylvestre saponin rich fraction (GSSRF) by in-vitro on breast cancer cell lines (MCF-7 and MDA-MB-468) and in-vivo efficacy in tumor bearing Elrich Ascites Carcinoma induced liquid tumor and Daltons Lymphoma Ascites induced solid tumor in mouse models.

I have some major comments:

-A language check is needed

-Please consider colors of the figure bars. Are the strong colors necessary?

-Is the presentation of significant difference with “a” and “b” in the figures according to the common style?

-In the introduction, several sentences describing mutations does not seem to make sense in this paper.

-In section 3.4, the following statement is made:

“The saponin rich fraction was found to be relatively safe on normal Vero cell lines.6”

Firstly, are there really any “normal” cell lines?

Secondly, there are no results shown.

-Can the authors please clarify how the positive control (Diallyl disulfide-1mM) was used in the in vitro assays?

- For the results in Table 4, authors claim that “more cytotoxicity was observed with the double positive breast cancer cell line MCF-7 (Table 4).” Is there a statistically significant difference?

-In figure 6, the authors show necrotic changes in sections from tumors (DLA-induced mice). The only proof of antitumor activity is discussed in the Discussion section. The authors should add results showing whether the suggested apoptotic pathways are involved or not, otherwise they can not claim cell death.

Round 2

Reviewer 2 Report

This paper has been improved according to the suggestions of the reviewers. I guess the paper may be very close to its publication. But please let me suggest some more points.

#Abstract: About the unit, mg/kg: Now I can understand why the authors use mg/kg. I can understand it is kind of a custom to use mg/kg in the scientific field of animal experiments, right? But I am afraid the readers unfamiliar with this field (frankly speaking, I am afraid I am one of those) cannot understand why this is mg/kg. Thus I want to suggest the authors to describe, at least in the abstract, like be safe at 2000 mg/kg body weight a day or 100 and 200mg/kg a day. As one of unfamiliar ones, I would like this.

#Table 1, Evaluation: Percentage increases in body >>> Percentage increases in body weight (?)

#Histopathology: The authors response to my comments is understandable. Then, I want to suggest the authors to include (insert) the authors comments (just those described on page 5 of authors response letter: i.e. the comments of the authors in addition to the reference shown there) somewhere in the Ms (I, in the authors shoes, would describe those in the Materials and Methods section (2.11. Histopathology). In this way, the readers can understand why the authors targeted those areas (LNs, etc) in this study. And one more: the HE photos in the revision are okay; although they are not the best ones, it is better for the readers to get a glimpse at the histology in the Ms.

#The results (data, tables or figures), on the whole, seem to be more convincing and flawless, seemingly increasing the value of the paper. But I want the authors to check the whole of these again and again.

#The grammar has been improved considerably in the revision. But still, I can find some questionable points about the grammar. I want the authors to carefully check the Ms again if the paper is to be published and to be read worldwide.

-Throughout the Ms: anti-oxidant or antioxidant? Anti-tumor or antitumor? Similar points, etc, etc, etc.

(If a word represents the same thing, it is better to find it to be uniform throughout the Ms)

-abstract: Acute Toxicity study of GSSRF  >>> Acute toxicity study of GSSRF (?)

-abstract, last 4th line: the Cisplatin a standard >>> the Cisplatin, a standard

-etc, throughout the Ms
